# A Review on the Technological Advances and Future Perspectives of Axon Guidance and Regeneration in Peripheral Nerve Repair

**DOI:** 10.3390/bioengineering9100562

**Published:** 2022-10-17

**Authors:** Arjun Prasad Tiwari, Taylor Lokai, Bayne Albin, In Hong Yang

**Affiliations:** Mechanical Engineering and Engineering Science, The University of North Carolina at Charlotte, Charlotte, NC 28223, USA

**Keywords:** axon guidance, peripheral nerves regeneration, bioengineering approach, optogenetic stimulation

## Abstract

Despite a significant advance in the pathophysiological understanding of peripheral nerve damage, the successful treatment of large nerve defects remains an unmet medical need. In this article, axon growth guidance for peripheral nerve regeneration was systematically reviewed and discussed mainly from the engineering perspective. In addition, the common approaches to surgery, bioengineering approaches to emerging technologies such as optogenetic stimulation and magnetic stimulation for functional recovery were discussed, along with their pros and cons. Additionally, clear future perspectives of axon guidance and nerve regeneration were addressed.

## 1. Introduction

Traumatic peripheral nerve injuries are common due to increasing traffic accidents, gunshots, tumor infiltration, electrical injuries, sports, and industrial accidents. Annually, more than a quarter million people, in the United States alone, suffer from peripheral nerve injury, resulting in loss of nerve function and compromised quality of life [1]. Functional impairment of the peripheral organs, due to nerve defects, results in multiple negative impacts. These hindrances can include those of personal lifestyle, function, and work, and will eventually increase social and economic burden on the healthcare system. Nonetheless, peripheral nerve tissues having regenerative capability, unlike central nerve fibers, which is a silver lining.

Axons are a fundamental unit of peripheral nerves, which originate from the base dorsal root ganglion (DRG) within the spinal cord region. Peripheral nerves are characterized by carrying the sensory electrical signals throughout the body. Peripheral axons exist in both myelinated and unmyelinated forms, where myelination is an axon covering made up of a layer of fat created from Schwann cells. In addition to myelination, axons are surrounded by three connective tissue sheaths, which support and protect both the axons and myelin sheaths. The innermost, surrounding axon sheaths at the individual level are called endoneurium, while the bundles of the axons along with myelin sheaths, called fascicles, are surrounded by perineurium (Figure 1). A third layer covers the entire peripheral nerve and protects the axons from surrounding tissues and helps prevent stretching.

The morphology of peripheral nerves characterizes axons to extend up to meters. The elongated axons, which are located farther from the cell bodies, are susceptible to potential mechanical injuries and subsequent nerve defects. The axons have the terminals, which are selective for substrate recognition, called growth cones. Once the nerve gap is formed, the growth cones fulfill the role in substrate recognition, physically, chemically, and precisely reaching their targets. Several factors such as the location of the defect, and the degree of local damages, influence the axonal guidance and growth [3]. The effect of local damage needs additional intervention to restore pre-injury existence with high accuracy in a reasonable time. For instance, the use of biomaterial grafts to promote functional recovery, through axon guidance and regeneration, has been a recent potential consideration [4,5]. Apart from nerve grafting and biomaterial usage, stimulation of nerves using magnetic and electrical field generation, along with chemical and optical stimulation are also considered for axon guidance and regeneration. In this review, we discuss the recent advances in experimental strategies, which have been studied for axonal guidance following nerve injury, their limitations, and future outlooks.

## 2. Strategies Adopted for Peripheral Nerve Regeneration

Different axonal reinnervation approaches have been widely reported. Major approaches include a surgical approach; a biochemical mediated approach; a biomaterial approach; an electrical stimulation approach; an optogenetic stimulation; and a magnetic stimulation approach.

### 2.1. Surgical Approach

The degree of peripheral axon damage varies from severe, major loss of function to mild, with some sensory and motor function deficits. Surgical intervention is required depending on the severity [6]. However, Sunderland grading is more frequently used by surgeons to distinguish when and how the injury is intervened [7]. They currently rely on the high-resolution and high soft tissue contrast magnetic resonance imaging, due to its increasing availability and high precision detection before intervention [8]. According to Sunderland grading, the first degree of injury characterized by mild loss of function is related to dysfunction resulting from compression, blocked blood flow, and loss of conduction, without the physiological damage to the axons [9]. Axonotmesis is a second-degree injury, characterized by only axonal damage, while the distal architect and myelin remain intact. These first and second degrees of injuries are left to heal on their own. However, third to sixth degree nerve injuries require surgical intervention. The third degree is characterized by the disruption of myelin, and glial scar formation in the endoneurium. The fourth degree includes perineurium disruption and nerve malfunction. A complete transection of the epineurium and corresponding connective tissue categorizes the fifth degree, and mixed consequences of injuries from first to fifth degrees are encompassed into the sixth degree, needing surgical repair to restore regeneration [10].

End-to-end suture has been a current gold standard in nerve reconstruction if the gap is less than one centimeter in length, and tensionless [11]. If reconstruction is not achievable by end-to-end suture, autologous nerve graft (which is harvested from the donor site and sutured to bridge the defect gap) has been considered the gold standard for managing nerve defects in a clinical practice. However, the functional outcome of autografting more than 3 cm in length is often poor [12]. In one study, a rat sciatic nerve defect model was used to compare the regeneration within short (2 cm) and long (6 cm) autografts [4]. They found that long nerve grafts had a reduced number of regenerated fibers and motor neurons compared with short nerve grafts [3]. The major limitation of this practice is the loss of sensitivity of the donor site and limited availability of autograft donor tissues [3]. Artificial nerve grafts are being used as an alternatives in the case of autologous donor tissue shortages.

### 2.2. Biochemical Approach

Direction and extension of axonal growth cones are important to meet the target point and functional repair. Initiation of this direction stems from a variety of in-built chemical cues that steer the growth cone through the chemotaxis. Extracellular matrix components such as ephrins, neurotrophic factors, and other biochemicals influence cell-to-cell contact and cell-substrate communication. Moreover, the chemical gradients have been shown to affect neurite outgrowth [4]. Individual axons create the neuronal network and are destined for the targets. Axon growth cones respond to guidance cues through the interactions with specialized receptor complexes. For example, integrin, ephrins, netrins, semaphorins, and slits instruct the axons to bind the Eph receptors, netrin (DCC and UNCS), neuropilins, and plexins, respectively [13,14]. Besides these classical axon guidance proteins, lipids are also important guidance molecules.

Calcitonin gene-related peptide (CGRP), an anti-inflammatory neuropeptide, is reported to increase fibroblast motility and extracellular matrix synthesis, vascularization, and proliferating Schwann cells. These features contribute to gaining peripheral nerve repair [13]. In another study, fractalkine, a chemokine, was embedded into alginate gel, and then introduced into nerve gaps [15]. Results showed embedding of the fractalkine enhanced axonal regeneration and muscle reinnervations. The results were comparable to the autograft, which was attributed to the recruitment of the reparative monocyte in the site which is proangiogenic and anti-inflammatory. Type III neuregulin 1 regulates pathfinding, the axonal survival of DRG neurons in the developing spinal cord, and peripheral injuries [14]. A number of studies have shown that ciliary neurotrophic factor (CNTF) promotes the axonal formation, survival, and regeneration in in-vitro culture of DRGs. Further enhancement of the neurite growth was observed when brain-derived neurotrophic factor (BDNF) was added [16]. In another study, increased levels of insulin-like growth factors had also been shown to enhance peripheral motor axons sprouting [17]. Moreover, glial cells also play the role of axon guidance; their positioning is key. Schwann cells secrete both nerve and fibroblast growth factors to maintain the regenerative microenvironment for axonal elongation and sprouting after traumatic injuries [18]. Therefore, the integrity of the Schwann cells is an important factor for enhancing axon growth [19]. However, the damaged axons are not well positioned to favor the desired biochemical pathways by themselves. The simple administration of the signaling molecules into the injured area is not sufficient to restructure the neuronal tissues due to their short half-life and non-specific delivery. Moreover, the gradient and concentration of these molecules in the repair site have to be maintained for successful regeneration and functional outcome.

### 2.3. Biomaterial Approach

Engineered tissue constructs have already been staged in clinics as a bridge of a peripheral nerve gap, while expecting to overcome the limitations and damage caused by nerve transfer and nerve grafting. Great attention has been given to the development of hollow nerve conduits as an alternative to autografts due to their biocompatibility, biodegradability, low cost, simple fabrication, and scalability [5,20]. The synthetic nerve conduit materials are fabricated from polymers such as, polyethylene terephthalate [21], and poly(L-lactide) [22], poly(ε-caprolactone) [23], poly(lactic glycolic acid), and collagen [24].

Currently, hollow nerve conduits are FDA approved to treat nerve defects less than 3 cm [25]. The production of hollow tube conduits, along with the interior lumen wall, and sheets of the aligned nanofibers are increasingly used. Such structures selectively house the axons, retract the fibrous tissue infiltration, allow axons to grow, and reduce neuroma and scar formation [26]. The regenerative process in nerve conduits is completed in three stages [27]. The first stage is the fluid stage in which the infusion of plasma exudate (fibrinogen and factor III) from the proximal and distal stump comes. This initial stage is followed by the formation of an acellular fibrin cable in the gap; this is called the matrix stage and is completed in the first week of repair, in contrast to 3 to 4 h in the first fluid stage. The third stage is called the migration phase, where the fibroblast’s endothelial cells migrate along the fibrin cable that forms in the matrix phase. Moreover, the Schwann cells subsequently proliferate, align, and form the Schwann cells cable, i.e., the glial bands of Büngner, where the axonal phase of repair is completed. At this stage, the new sprouts observe, navigate by individual growth cones, and ultimately reach their targets. Later Schwann cells wrap the bare axons and transform into myelinated axons. This whole process usually occurs in 4–16 weeks [28]. Still, functional recovery is hardly achieved. Failing of the axon regeneration is mainly due to the poorly assembled and insufficiently bridged fibrin network in the matrix phase. To overcome this, several modifications of the hollow-based conduits have been reported to provide additional cues and topographical guidance [29,30].

Intraluminal guidance structures and micro-grooved luminal designs provide additional support to the fibrin matrix, guiding the myelinating cells and regenerating axons [31]. The packing density of the intraluminal supporting structure greatly influences the recovery process. For example, embedding high-density polylactide microfilament into the luminal region inhibited nerve regeneration [32]. Electrospinning scaffolds have been widely used as axon guidance and support. These provide the initial adhesion and guidance by mimicking the nature of the cellular microenvironment, having tunable porosity, and serving as a template for the axon’s growth [33,34]. However, inability to support three-dimensional growth of the cells limits uses in real settings. Recently, combination approaches have been considered promising to provide improved axon regenerative properties. Chew et al. [35] fabricated a biodegradable, biofunctionalized, three-dimensional-aligned nanofiber–hydrogel construct for spinal cord injury treatment. The scaffold comprises the aligned polycaprolactone (PCL)-co-ethyl ethylene phosphate nanofibers dispersed in the collagen hydrogels. The collagen largely mimics the extracellular matrix protein, while aligned nanofibers mechanically support axons and direct the neurite extension. Such scaffolds were found to enhance axon regeneration and remyelination in an in vivo rat model.

Still, the nerve conduits are inferior to the autografts because they lack the necessary support for the regeneration and functional cell binding clues. They are not sufficient to direct axon growth and further maturation. Patterning of the laminin (as a putative axon adhesion and guidance molecule) on chitosan scaffolds promotes a DRG axon sprouting, preferentially grown on the pattern [1]. Immobilization of pro-regenerative biomolecules to culture substrates has been utilized in neural guidance during development and injury. An RGDA peptide and axontin-1, cell adhesion protein was immobilized to a substrate, which showed extensive neurite growth and network formation [32]. Despite FDA approval of nerve grafts used to reconstruct the defect, no grafts have been approved for a gap exceeding 3 cm or longer. In real world medical practice, long nerve grafts are needed, especially for multiple extended injuries such as plexus nerves by trauma or long length tumor infiltration peripheral nerve tissue. Therefore, developing a long nerve conduit exceeding 3 cm which can demonstrate sufficient nerve regeneration is critical. The representatives of nerve conduits are shown in Table 1.

### 2.4. Electrical Stimulation Approach

The clinically relevant electrical-stimulation approach enhances the intrinsic regenerative capacity of neurons. The studies on the peripheral nervous system strongly suggest the advantages of electrical stimulation on sensory and motor neuron regeneration [43,44]. In one study, the increased neurite growth was found in chick embryonic DRG cells under an electric field [44]. The enhanced peripheral neuronal growth is attributed to the upregulation of the nerve growth-associated genes (GAP-43) [45], neurotrophic factors, and BDNFs [46] and glial cell line-derived neurotrophic factor (GDNF) [47] in DRGs. The duration and power of electrical stimulation are the factors affecting regenerative ability. However, the optimal physical factors cannot yet be predicted and can be dependent on each case.

Verge group [44] studied the effect of electrical stimulation on regeneration in a nerve gap of 20 mm in a rat model. The cathode was sutured alongside the femoral nerve, just below its exit from the peritoneal cavity, whereas the anode was sutured to muscle more distally, close to the nerve and just proximal to the suture repair site. The wires were connected to a custom-made biocompatible implantable stimulator that was encased in epoxy resin and covered with biocompatible silastic and contained a light-sensitive diode, which turned the stimulator on and off by an external light flash. Stimulation commenced immediately after nerve repair with supramaximal pulses (100 μs; 3 V) delivered in a continuous 20 Hz strain by the implantable stimulator. They found that alternative current electrical stimulation enhanced neuronal regeneration [48]. This was correlated with the increased expression of corresponding biochemical cues. Singh et al. [49] tested electrical stimulation at the proximal to transected sciatic nerves in the mice. Electrical stimulation resulted in 30–50% improvement in several indices of the axon regeneration, such as regrowth of axons and bonding of their partnered Schwann cells across the transection sites, developing neuromuscular junctions. The mouse model studies were further supported with in vitro studies in which accelerated neurite outgrowth was found. It is noteworthy that stimulation at a lower frequency led to a superior regeneration of sciatic nerves, compared with groups receiving a higher frequency [50]. In that study, a 10 mm nerve gap was made, and sutured stumps into silicon rubber, followed by stimulation at various frequencies. Uses of 2 Hz had higher axon density, more myelinated fibers, and a higher ratio of blood vessels compared with control in a rat model. In the same study, electrophysiology assays showed higher conduction velocity and shorter latency when used at low frequency.

Further advancing the axon regeneration research, Yang et al. [51] showed that co-cultures of DRGs neurons and Schwann cells into the microfluidic chamber exhibited improved myelination (Figure 2); showing a fivefold increase of the myelinated segments when compared to the non-stimulated samples. Axon sprouting and alignment were observed in the second chambers that connected to the first chamber, where the cell body remains followed by selectively stimulation of the axons; 10 Hz pulses at a constant 3 V (with 190 W impedance) were employed for the stimulation. This approach of compartmentalized chamber usage stimulates axons precisely in natural conditions, where the cell bodies are far from the injured axon site. Still, there are several limitations of electrical stimulation such as poor biocompatibility and reduced ability for prolonging implantation. Surrounding muscle fibers and connective tissues can be damaged [52]. The operational procedure of electrodes to manage mechanical proximity to tissue and electrical integrity can be difficult. The precise parameters for systematic stimulation, while avoiding overexcitation, are also still lacking.

### 2.5. Optogenetic Stimulation

Optogenetic stimulation has emerged as a potent tool in neuroscience engineering. It is a non-invasive procedure and has high selectivity, which may outweigh other counterparts’ stimulation techniques. The data from the different groups showed that optogenetic stimulation promotes neurite outgrowth [53,54]. The optical pulses and exposure time are influencing factors for the neurite outgrowth and axonal regeneration [53]. Park et al. [55] explored optogenetics as a means to promote neurite growth taking light-sensitive whole DRGs from transgenic Thy1-ChR2-YFP mice expressing ChR228 with the hypothesis that optically induced neural activity will increase neurite outgrowth. They studied a various range of optical stimulation frequencies and exposure durations on the outgrowth of neurons. Additionally, they found increased and directionally biased outgrowth of optically sensitive neurites, exemplifying the cell-specific targeting of optogenetics.

Selective axons are subjected to advanced optogenetic stimulation-enhanced, activity-dependent myelination. For example, DRGs neurons with an expression of light-sensitive protein channelrhodopsin 2 (ChR2) were co-cultured with Schwann cells in a compartmentalized chamber consisting of axonal and soma chambers, connected through the channels [53]. The neuron cells are highly polarized, where the axons are in distinct stems from the cell bodies and run directionally. A 50 mW blue light-emitting diode of 470 nm wavelength flashing 0.5 s in 2 s intervals was exposed to co-culture in the axonal chamber. Results exhibited enhanced axon regeneration, myelination, and oligodendrocytes differentiation promotion. This study predicted that axon stimulation is sufficient to increase neuronal activity induction in peripheral tissues. This approach alone may not be sufficient to induce regeneration in a high degree of injuries; however, it emphasizes the potential to use optogenetic stimulation in combination with other approaches stated. Considering these benefits, optogenetics can be a potential tool for retaining the functional recovery following peripheral nervous system injury.

### 2.6. Electromagnetic Stimulation

Magnetic stimulation is a prominent noninvasive method used to stimulate neuronal growth [56]. Although electrical stimulation is the most common approach for neuromodulation, several limitations come along with it. This includes the invasiveness, and detrimental effects on the electrode performance for long-term usage. In comparison with electrical stimulation, magnetic stimulation of peripheral nerves does not attenuate the performance over time [57]. Magnetic stimulation uses a similar technique to electrical stimulation of applying an electrical field in pulses to induce outgrowth, fortunately without the invasive drawbacks of electrical stimulation. Several in vitro studies on the neuron cells strongly support that remotely controlled magnetic stimulation potentiates the outgrowth of the neurites and interaction among the neurons. Gilbert et al. showed that iron oxide nanoparticles, which are embedded nanofibers, showed enhanced neuronal outgrowth in response to magnetic stimulation compared with non-stimulated samples [58]. To demonstrate this claim, particles were grafted into the nanofibers during electrospinning. Their study emphasizes that the aligned nanofibers, with activation by magnetic field, have higher potential for nerve guidance compared with the corresponding individual counterparts. The reason for the improved neuronal response to magnetic stimulation is thought to be that charged particles induce the mechanical tension in and around the cell, and that may play the role in a mechanistic way. Weak, static magnetic fields were found to be neuroprotective against anticancer drug, etoposide-treated primary neuron cells, which are under prolonged survival and reduced apoptosis in a time-and-dose-dependent manner [59]. Protection by static magnetic field was attributed to the altered Ca^2+^ flux through voltage-gated channels. Enhanced axon growth and differentiation capacity of oligodendrocytes into Schwann cells, following a static magnetic field of 0.3 T for 2 weeks (two hours/day), were observed in an in vitro study of a microfluidic device. The co-culture of axons with oligodendrocytes expressed the significantly pro-myelination gene marker c-fos, early OPC (Olig1, Olig2, Sox10) [60].

Further advancing the magnetic stimulation approach, transcranial magnetic stimulation (TMS) has been used as a non-invasive method for the brain or injured spinal cord stimulation, which is electromagnetic induction using an insulated coil, placed over the injured site. This is believed to increase neural activities, leading to increased neural regeneration [61]. The metal coil produces magnetic pulses, which pass through the barrier into the site easily and painlessly. The frequency and pulses generated are of a similar type and strength to those produced by magnetic resonance imaging. Moreover, this technique reduces inflammation and lesions in addition to increasing angiogenesis [62]. In one study, a wireless stimulator based on a metal loop powered by a TMS without circuitry components was proposed. A loop was embedded into the chitosan and bonded with sciatic nerves with laser assistance. Results revealed that axon regeneration was observed in the area of the transection site when stimulated 1 h/week. TMS induced high compound action potentials in muscles and nerves, whereas there was no action potential elicited in TMS stimulation without a loop. This highlights the necessity of the combination of the approaches in virtue of functional recovery [63].

A number of studies have shown that low-frequency pulsed magnetic fields increase neurite outgrowth by altering the ion channel functions along with increasing nerve conductivity and action potentials [64,65]. Due to low impedance in the wire coil, the generation of the magnetic field with low frequencies consumes a substantially high amount of energy. As a result, sub-micrometer implantable magnetic coils are paired with bulky power sources. With this reasoning, clinical usability is limited, despite the great potential. Engineering a platform which delivers low-frequency magnetic pulses via a miniature low-powered stimulator can open doors to the exploration of the nervous system and beyond, via wearable devices.

## 3. Concluding Remarks and Future Perspectives

Autografts are still superior to any bioengineered grafts for nerve reinnervation. However, the resulting negative limitations anticipate the development of alternative approaches. On the other hand, chemical cues for the cell-to-matrix or cell–cell interactions are required to achieve the regenerating ability of neurons. Nonetheless, their complexity and biological nature show a low half-life; these are not sufficient as a single treatment to increase axon guidance and regeneration. The combinations of bioengineered grafts with signal molecules were further studied to match or exceed the autografts model. Most biomaterial approaches were found to focus on only the development of nerve conduits facilitating nerve guidance and growth. However, negative consequences such as compression of nerves and nerve/muscle tension induced in the local microenvironment, following axon guidance, have not been considered with much attention. In other words, the possible benefits of the axon-guided materials, controllably removed once they perform as the guidance material, have not been investigated. Therefore, developing on-demand degradable, axon-guided scaffolds with signaling molecules and assessing them further to identify the advantages over conventional conduits can be future endeavors.

Apart from those mentioned, electrical stimulation, optogenetic stimulation, and magnetic stimulation are becoming promising tools for peripheral nerve regeneration. Electrical stimulation is a potential approach to that. However, the hindrance to succeeding therapy are low biocompatibility and problems at the electrode–tissue interfaces during long-period implantation [66]. Implanted materials for electrical stimulation placed under the cover of insulated biocompatible materials, such as miniature coils, offer several advantages in biocompatibility and operational feasibility, and should be addressed. Further noninvasive approaches are also considered potential therapies. Magnetic stimulation has been widely studied for the non-invasive treatment of neuron malfunctions. Nonetheless, a detailed mechanistic and molecular approach to how the neurons benefitted from said stimulation is not understood. The challenge of using magnetic field stimulation now stems from the miniaturization of the device and optimizing the pulse. Source-generating, low-frequency pulses in miniaturized coils can only be generated by large amplifiers. An additional obstacle is the duration of the stimulation pulses. Therefore, lowering power to induce a low-frequency pulse may be realized for future consideration. Moreover, magnetic stimulation of three-dimensional cellular constructs (such as organoids) would be a promising choice for neural stimulation. Electrical and magnetic stimulation approaches can be debatable regarding the effector cells or subcellular organelles upon therapy. Therefore, the subcellular compartmentalization of neuron cells, followed by stimulation by different approaches, would further increase the suitability of the approaches.

## Figures and Tables

**Figure 1 bioengineering-09-00562-f001:**
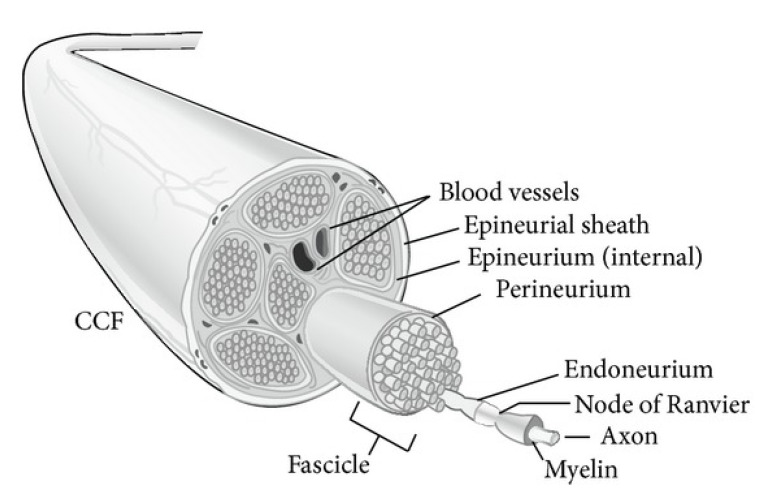
Peripheral nerve (axon) anatomy. Reproduced with permission from ref. [2].

**Figure 2 bioengineering-09-00562-f002:**
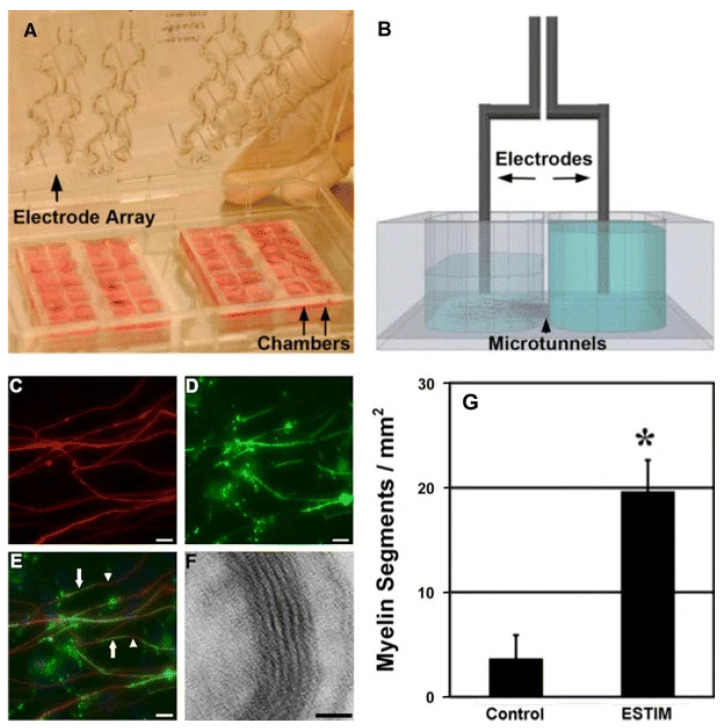
Compartmentalized microfluidic device consisting of two chambers connected with parallel channels. The chambers are inserted into a tray fitted with a lid containing 4 arrays of 5 electrode pairs used for stimulating both chambers at the same time (**A**), scheme showing a connection of electrodes in a chamber (**B**), myelination study; MBP-positive oligodendrocyte processes (green) contact with neurofilament-positive DRG processes (red) in the distal compartment of the microfluidic platform after 5 days of co-culture and stimulation. Arrows indicate MBP-positive myelin formation and arrowheads indicate non-myelinated axons. Scale bars = 15 μm (**C**–**E**), TEM image of myelin formation, 6 to 10 wraps were formed. Scale bar = 20 nm (**F**) and comparison of myelin segment formation between control groups and electrical stimulated groups. Fivefold more myelin segments were observed in the group receiving electrical stimulation than control group (Asterisk indicates a *p* < 0.005). (**G**) (permission taken [51]).

**Table 1 bioengineering-09-00562-t001:** Nerve conduits used in peripheral nerve guidance and regeneration.

Materials	Scaffold Type/Fabrication Technique	Key Results	Ref.
Polyacrylic acid (PAA) polyamidoamines	Hydrogel tubing/polymerization	Improved sciatic nerve regeneration, no inflammation	[20]
Polycaprolactone/Polydimethylsiloxane (PCL/PDMS)	Nanofibers-microfludic device/electrospinning-microfabrication	Improved axon guidance and myelination	[36]
PCL/PDMS	PCL coating on PDMS/Spin coating	Micro topographical cues improve nerve regeneration	[37]
Chitin/polydopamine	Hollow chitin hydrogel tube/freeze-thaw method	Inhibit neuroma formation	[38]
PCL-based	Hollow conduit (made by Neurolac)	Improved functional recovery	[5]
Polyurethane-carbon nanotube	Conductive Align nanofibers	Increased neuron cells aligned, differentiation and regeneration	[39]
Deendothelialized nerve conduit	Nerve tube/cellular manipulation	Motor recovery function compared with autograft, increased vasculaization	[40]
Polylactide	Microporous conduit/solvent-non-solvent phase conversion	nerve bundles formed and long-term support, achieving a functional recovery	[41]
Cell encapsulated-gelatin methacrylate (GelMA) and poly(ethyleneglycol)diacrylate (PEGDA)	3D printing	Platelet encapsulation leading to the sustained release of multiple growth factors, platelets significantly promoted the hydrogel conduits in peripheral nerve repair in vivo	[42]

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
