# Peer review of "A Review on the Technological Advances and Future Perspectives of Axon Guidance and Regeneration in Peripheral Nerve Repair"

_bioengineering, 2022, doi:10.3390/bioengineering9100562_

Round 1

Reviewer 1 Report

In this paper, the authors have reviewed most of the current technologies used for axon regeneration with the pros and cons of each. Moreover, they have discussed the possible future works for enhanced axon regeneration. The issue has higher scientific and clinical significance. The manuscript is well-written in terms of content-wise and English-wise. Overall, I suggest publishing this after minor modifications.

The other comments are:

1.       The authors only mentioned 7 relevant works in table 1. More contemporary other works can be added to table 1.

2.       The graphical abstract in ref 7 of table 1 is missed. Please add.                 

3.       The discussion about the limitations of current magnetic stimulation approach is not clearly mentioned, please revise section 2F.

4.       I found some abbreviations such as BDNF, and GDNF are mentioned without full form. Need to be addressed.

5.       I think some relevant references should be added. I found shortages of refs in many contexts.

6.       I also suggest providing graphical abstract to depict their work more clearly.

Author Response

Response to reviewers comments

We are thankful to the reviewers for valuable comments. We have revised the manuscript according to your suggestions. To make further review easy, revised texts were marked with blue color text.      

Reviewer 1

In this paper, the authors have reviewed most of the current technologies used for axon regeneration with the pros and cons of each. Moreover, they have discussed the possible future works for enhanced axon regeneration. The issue has higher scientific and clinical significance. The manuscript is well-written in terms of content-wise and English-wise. Overall, I suggest publishing this after minor modifications. The other comments are:

Our response: Thank you very much for your positive feedback. We try our best to improve the manuscript by addressing all       your concerns. 

  1. The authors only mentioned 7 relevant works in table 1. More contemporary other works can be added to table 1.

Our response: We have revised the table by adding other relevant work Ref, 37 and 38. Please see       Table 1.

  1. The graphical abstract in ref 7 of table 1 is missed. Please add.                 

Our response: Thank you for your suggestion. There is no graphical abstract provided of ref 7.

  1. The discussion about the limitations of current magnetic stimulation approach is not clearly mentioned, please revise section 2F.

Our response: Thank you so much for your concerns. We have revised the texts accordingly. Please see       the highlighted texts on section magnetic stimulation.

  1. I found some abbreviations such as BDNF, and GDNF are mentioned without full form. Need to be addressed.

Our response: We have Addressed the concerns and mentioned the full form in the revised manuscript.

  1. I think some relevant references should be added. I found shortages of refs in many contexts.

Our response: Thank you so much for your concerns. We have added new references in the revised manuscript.

  1. I also suggest providing graphical abstract to depict their work more clearly.

Our response: Thank you for your suggestion. We have added a new graphical abstract section. This may appear along with the Title and     Abstract, if accepted      in the “Bioengineering” journal.    

Reviewer 2

In general, this is an interesting review by the authors or various techniques to promote neural regeneration.  I think it would benefit from some editing regarding word choice and sentence structure. I also think they would benefit from having a surgeon edit and revise some of the sections regarding surgical approaches. Their conclusions in those sections are somewhat flawed and require substantial revision. Specific comments below, although overall I thought the paper was very informative.

Our response: Thank you very much for your positive feedback and most sincere concerns. We did       our best to address the comments you raised. 

Line 28: Reword sentence – awkward

Our response: The introduction is revised. Sentences are reshuffled completely. Some are removed or edited. Please see introduction section.

Line 31: This doesn’t make sense

Our response:       This sentence      has been removed, and      the whole introduction has been revised.

Line 32-34: This is not accurate, we use autografts for much larger defects, and in older patients, and the autograft does not have a risk of tension if done correctly.

Our response: Thank you so much for sharing your thoughts regarding the autograft. We have removed those lines from the introduction section and included concerns further in the surgical section, which has been      revised greatly. Please see 60-88.

 47: What is a mechanical trigger?

Our response: Thank you very much for your concern. We remove the words “mechanical trigger” to avoid possible confusion.

Line 57 -59: The problem is much more complex than just trying to regain function more quickly. The real problem is the lack of accurate and complete neural regeneration, not just the time.

Our response: We agree with the reviewer on your insightful thoughts on the theme of the mentioned lines. We meant to say the uses of additional therapy or tools can help to restore functional outcome with high accuracy at reasonable time. We have further revised the sentence. Please see lines 46-50.

Line 74: I don’t believe injural is a word

Our response: We have removed the word and revised the sentence. Please see line 61.

Line 88-95: This does not accurately reflect the surgical strategies, benefits or limitations thereof

Our response: Thank you so much for your critique, we have revised the whole surgical section.         

Line 138: We don’t use conduits for large defects, only small defects and even then they have fallen out of favor.

Our response: Thank you so much for your sharing experiences. We have revised the section. Please see lines 172-182.

Your concern       regarding nerve conduits      falling out of favor is a critical view to be discussed in future works.

Reviewer 3

This review article highlights the current progress in peripheral nerve regeneration, with a

particular focus on axon guidance. The authors provide an overview of some approaches being

developed toward promoting outgrowth and navigating axons following peripheral nerve damage.

Although the authors provided an informative overview of the topic, there is a substantial lack of

necessary details to help readers to track the research progress within a specific direction. More

descriptive information would be helpful to add.

Our response: Thank you very much for your positive feedback. We revised the manuscript substantially by addressing reviewers’ concerns.

- Also, there are some inconsistencies throughout the text. For example, the authors started the first paragraph of the introduction by stating the ability of peripheral nerves to regenerate, whereas they claimed unsatisfactory outcomes shortly after that. More work is needed to maintain the logical flow of the whole text. Similarly, “the abundantly-present growth factors following injury” (list those!) should logically help the regeneration process (lines 28-29).

Our response: The introduction section is revised. The mentioned inconsistencies are corrected.

- The surgical approach is the most clinically used at present; therefore, authors have to put more

effort for better clarity. What is a way of evaluating the degrees of nerve injury before deciding on

surgical intervention? It is a crucial point for this approach.

Our response: Thank you so much for critical concern. We have revised the surgical section substantially.

- None of the described approaches outlined the existing sporadic axonal sprouting, which is an

overwhelming problem requiring special attention with either approach.

Our response: Thank you so much for your important concern. We included the specific discussion about axonal sprouting in the revised manuscript. Please see lines 112-114, 171-173 etc.

- All abbreviations should be given with their full name first, including figures.

Our response: We have revised this      accordingly, throughout the manuscript.

- It is not convenient to find out what is shown in the snapshots in Figure 2. It must be improved.

Our response: Thank you so much for your comment.       We have used this Figure from the published report after getting permission to reprint. The caption is revised.

Reviewer 2 Report

In general this is an interesting review by the authors or various techniques to promote neural regeneration.  I think it would benefit from some editing regarding word choice and sentence structure. I also think they would benefit from having a surgeon edit and revise some of the sections regarding surgical approaches. Their conclusions in those sections are somewhat flawed and require substantial revision. Specific comments below, although overall I thought the paper was very informative.

Line 28 : Reword sentence – awkward

Line 31: This doesn’t make sense

Line 32-34: This is not accurate, we use autografts for much larger defects, and in older patients, and the autograft does not have a risk of tension if done correctly.

Line 47: What is a mechanical trigger?

Line 57 -59: The problem is much more complex than just trying to regain function more quickly. The real problem is the lack of accurate and complete neural regeneration, not just the time.

Line 74: I don’t believe injural is a word

Line 88-95: This does not accurately reflect the surgical strategies, benefits or limitations thereof

Line 138: We don’t use conduits for large defects, only small defects and even then they have fallen out of favor.

Author Response

(The authors gave the same response as above.)

Reviewer 3 Report

This review article highlights the current progress in peripheral nerve regeneration, with a particular focus on axon guidance. The authors provide an overview of some approaches being developed toward promoting outgrowth and navigating axons following peripheral nerve damage. Although the authors provided an informative overview of the topic, there is a substantial lack of necessary details to help readers to track the research progress within a specific direction. More descriptive information would be helpful to add.

- Also, there are some inconsistencies throughout the text. For example, the authors started the first paragraph of the introduction by stating the ability of peripheral nerves to regenerate, whereas they claimed unsatisfactory outcomes shortly after that. More work is needed to maintain the logical flow of the whole text. Similarly, “the abundantly-present growth factors following injury” (list those!) should logically help the regeneration process (lines 28-29).

- The surgical approach is the most clinically used at present; therefore, authors have to put more effort for better clarity. What is a way of evaluating the degrees of nerve injury before deciding on surgical intervention? It is a crucial point for this approach.

- None of the described approaches outlined the existing sporadic axonal sprouting, which is an overwhelming problem requiring special attention with either approach.

-  All abbreviations should be given with their full name first, including figures.

- It is not convenient to find out what is shown in the snapshots in Figure 2. It must be improved. 

Author Response

(The authors gave the same response as above.)

Round 2

Reviewer 3 Report

A thorough English check is still required.

Author Response

Response to reviewers’ comments

We are thankful to the reviewers for their valuable comments. We revised the manuscript by extensive English corrections by native English speakers. The coauthors Taylor Lokai and Bayne Albin corrected. Thank you.